# Traditional Knowledge Holders and Practitioners: First Responders in Native Nations During the COVID-19 Pandemic

**DOI:** 10.3390/ijerph22091432

**Published:** 2025-09-14

**Authors:** Nicolette I. Teufel-Shone, Amanda Hunter, Carol Goldtooth-Begay, Manley A. Begay, Andria B. Begay, Darold H. Joseph, Melinda S. Smith, Julie A. Baldwin

**Affiliations:** 1Center for Community Health and Engaged Research, Northern Arizona University, Flagstaff, AZ 86011, USA; carol.goldtooth@nau.edu (C.G.-B.); andria.begay@und.edu (A.B.B.); melinda.smith@nau.edu (M.S.S.); julie.baldwin@nau.edu (J.A.B.); 2College of Health Solutions, Arizona State University, Phoenix, AZ 85004, USA; amanda.urbina@asu.edu; 3Applied Indigenous Studies Department, Northern Arizona University, Flagstaff, AZ 86011, USA; manley.begay-jr@nau.edu; 4School of Medicine and Health Sciences, University of North Dakota, Grand Forks, ND 58202, USA; 5College of Education, Northern Arizona University, Flagstaff, AZ 86011, USA; darold.joseph@nau.edu

**Keywords:** Native American, COVID-19 pandemic, traditional knowledge holders

## Abstract

Native Americans in the US experienced disproportionate risks of COVID-19 infection and mortality. Despite these adversities, Native Americans relied on the world view and lessons of their cultural teachings, as strategies to find personal solace and social harmony amid the crisis. Traditional Knowledge Holders and Practitioners (TKHPs) reinforced these survival strategies and were essentially first responders. In 2021, 22 TKHPs from three Arizona Native nations were interviewed about their personal reflections and practice during the pandemic. A cross-Native nation analysis of the narratives revealed three determinants shaped the health of Native peoples in these communities: (1) relationships with all living beings and the natural environment, (2) the intersection of non-Indigenous and Indigenous health care systems, and (3) cultural continuity. TKHPs’ contributions to their communities’ physical, social, cultural, and spiritual health during the COVID-19 crisis elucidates the need to ensure their inclusion in public health emergency response plans. Their knowledge and practice are foundational assets in Native American communities, offering invaluable lessons to promote mental wellness and resilience. TKHPs’ approach to addressing pandemic-related challenges extended beyond the typical Western approaches to medicine, making them vital providers for current and future efforts in improving the health status of Native Americans.

## 1. Introduction

Traditional knowledge refers to the wisdom and scientific skills Native American people value and have used since the beginning of time to discover the dependable, repetitive, and tested way things work in the world [1]. Traditional Knowledge Holders and Practitioners (TKHPs), referred to in English as medicine men and women or spiritual leaders, have an essential role as keepers of traditional teachings and lessons learned from their collective experiences, their predecessors, and their relationships with other forms of life in their natural environment and the world and universe around them [2,3,4]. As such, TKHPs are advisors and healers, grounding their prayers, ceremonies, and advice in epistemological foundations situated within this relationship-oriented, balance-seeking worldview [5,6]. The concepts of balance and harmony within oneself, with family, and with the community (local and external communities) structures the choices and actions of Native American people within this relational universe. TKHP knowledge and practice provide guidance on restoring and sustaining balance amid disruption [5,7]. The COVID-19 pandemic brought loss, hardship, and chaos worldwide. For Native American nations, TKHPs were among the first responders, working on the frontlines to support health and recovery throughout the pandemic using their experience with traditional knowledge systems [8].

Since the start of the COVID-19 pandemic in 2020 [9], Native Americans experienced disproportionate risks of COVID-19 infection and mortality. In the United States (US), the national 2020 COVID-19 mortality rates for Native Americans were almost three times that for Whites [10]. In Arizona, Native Americans experienced a mortality rate from COVID-19 about eight or more times higher than the White population [10]. Poverty and preexisting co-morbidities limited access to care, adequate housing, and indoor plumbing, which, in turn, contributed to the COVID-19-related disparities in infection rates, survivorship, and mortality within Native American communities [10,11]. Despite these adversities, Native Americans were steadfast, relying on their strong tradition of prioritizing the collective, caring for each other, and adhering to their tribal governments’ strict isolation and curfew policies [12]. TKHPs were vital to these survival strategies and were essentially first responders. As feelings of uncertainty and panic unfolded during the initial months of the pandemic, Native American people sought TKHPs for comfort and support. TKHPs provided prayers and consultation by phone, Zoom^TM^, and text messages [13]. They conducted ceremonies using modified protocols, which included maintaining safe distance between participants and avoiding shared ceremonial instruments, food, and water [13].

Traditional practices and spirituality played an important role in maintaining the health and well-being in Native communities during the pandemic and are important driving factors in the Indigenous Determinants of Health (IDOH) framework. IDOH is an advancement of the Social Determinants of Health (SDOH) and applies a lens that considers the impact of marginalization and cultural assets on health determinants among Native people [4,14,15]. The SDOH are grouped into five domains: Economic Stability, Education Access and Quality, Health Care Access and Quality, Neighborhood and Built Environment, and Social and Cultural Context [15]. The expanded IDOH framework incorporates Native Peoples’ and nations’ distinct challenges and adaptive strategies that affect their physical, social, mental, and spiritual health and well-being. Drawing on the literature addressing IDOH [14,16] and the expertise of our team, three determinants that shape the health of Native Peoples in the US were identified, which were not adequately addressed by the SDOH descriptions. These determinants include: (1) relationships with all living beings and the natural environment, (2) the intersection of non-Indigenous and Indigenous health care systems, and (3) cultural continuity.

The decision to establish these three distinct determinants is grounded in the IDOH literature and documentation that Native Americans, as a people, have life experiences distinct from other US minority groups [17,18]. Complex factors defined by treaty agreements [17,18], sovereign rights, Supreme Court decisions, changes in US historical and contemporary policies governing the interface with Native nations, and the health risks associated with climate change have unique implications for the health and well-being of Indigenous Peoples who have a profound connection to the land, water, and natural environment. These IDOHs are not considered by the SDOH and, by extension, the goals and objectives of Healthy People 2030 [15], warranting consideration for their impact on health [3,19,20]. For example, the Health Care Access and Quality Healthy People 2030 objectives address the availability of clinics, hospitals, and providers to provide screening and treatment but do not address the discordance between Euro-American and Native American philosophies and approaches to health, wellness, and health care [21].

In 2020, during a time of uncertainty imposed by the pandemic, a multidisciplinary university-based team of 11 US Native Americans and 2 non-Native investigators partnered with three Arizona Native nations to document the role of IDOH in supporting Indigenous mental health and well-being, and in turn, resilience during the COVID-19 crisis. The intention in using the IDOH framework was to recognize the preexisting stressors, world view, and socio-cultural practices that shaped Native Peoples’ and nations’ COVID-19 experiences. This paper shares reflections from the TKHP group on the social and spiritual disruption caused by the COVID-19 pandemic and their work to restore harmony and balance for themselves and their patients and communities.

## 2. Methods

### 2.1. Partnership and Approvals

The members of the university-based team drew on their existing relationships with Native nations in Arizona and considered nations’ ongoing experiences during the pandemic. Contact was made with Native nation leadership (e.g., health departments, government entities, or Traditional Medicine Societies) to discuss interest in research designed to document the role of community and cultural strengths in weathering the pandemic. Once interest in a university-Native nation partnership was established, research approval was sought and granted from respective Tribal councils, institutional review board, or Cultural Preservation Office, as directed by tribal protocol [22]. Partnerships were established with large and small, rural and urban nations, representing the diversity of contexts for Native people in Arizona.

### 2.2. Approach

The university-Native nation partnership agreed that key informant interviews were a culturally acceptable approach to documenting the impact on mental health, well-being, and resilience in four specific groups, first responders, educators, TKHPs, and the substance abuse recovery community living and/or working in or near three Native nations in Arizona [4,22]. The work with the TKHPs is discussed in this paper.

### 2.3. Recruitment

A community researcher from each Native nation was hired as a vital member of the collaborative research team to aid in recruiting participants, scheduling interviews, translating interviews from the Native language into English, and providing overall expertise to ensure that methods and approaches were culturally congruent and respectful [22]. The community researchers completed the Collaborative Institutional Training Initiative (CITI) training [23] and were familiar with participant eligibility criteria [22]. In the US, Institutional Review Boards require that CITI is completed by all investigators and study personnel involved in research involving human subjects. Community researchers were equipped with laptops, a mobile hotspot device (if needed), and access to shared drives of consent forms and logs for participants’ payments. A university coordinator was consistently available to answer questions. Community researchers were an integral part of the team and worked in their community, meeting with the university contact at least once a week or more, if needed.

Members of the research team were citizens of two of the partnering Native nations. For the third Native nation, one Native American member of the team had a >6-year partnership with the nation. Drawing on these social and cultural connections with the partner nations, university and community researchers discussed and identified the TKHPs well known in the respective communities and extended an invitation to participate in the interviews. There is no formal designation or certification of TKHPs; they may range from elders who know the traditional stories, herbalists and practitioners who run healing ceremonies or offer prayers to guide the restoration of balance. This community led approach supported the validity of the sampling design. The community researchers extended the invitation, reviewed the Human Subject Consent documents with the potential interviewees, gained consent for the interview via a signature and coordinated the interview time with the interviewers and interviewee.

### 2.4. Interviews

The whole team developed and edited interview questions guided by IDOH over 2 to 3 months. The research team created five “core” questions to analyze themes across groups and group-specific questions to discern group-specific challenges and coping strategies [22]. Detailed descriptions of the interview methods and codebook development for qualitative analysis have been published previously [22]. Briefly, the research team conducted interviews with TKHPs from three Native nations in Arizona over 6 months (May–November 2021). Interviews were conducted virtually, via Zoom^TM^, or over the phone, lasting 45 min to 2 h. Interviews were recorded, transcribed verbatim, translated (if needed), and transcripts were checked for accuracy before analysis. After the interview, the TKHPs received a $25 gift card and gifted mountain smoke (traditional tobacco) and an arrowhead for their time.

### 2.5. Analysis

Transcribed and translated text were analyzed using NVivo^®^ 10 and 11 [24]. Three independent coders (all Native American) were assigned to the TKHP narratives. Each coder was responsible for individual transcript analysis with no overlap, so the team did not conduct interrater reliability analysis. However, all coders contributed to the development of the codebook, which included parent codes based on the IDOH. Additionally, the coding team working on all the subgroups’ narrative analyses met bi-weekly to ensure consistency in the coding process [22]. A second set of four coders, representing each research group, reviewed the coded quotes using the IDOH-specific codes to concur, validate, and/or identify and resolve coding differences. These resolutions occurred in the bi-weekly meetings. To further guide our analysis, we developed a diagram (Figure 1) to illustrate the concept of disharmony and restoring harmony through traditional knowledge and practice.

## 3. Results

Table 1 provides the demographic characteristics of the 22 TKHPs from the three partnering Native nations who completed the interviews. Males composed > 70% of the total number of interviewees, which is reflective of the TKHP composition in many Native communities in North America [25].

Guided by the commonality of TKHPs’ roles to identify or diagnose disharmony and strive for balance and harmony, we identified themes of disruption and restoration of harmony through the recognition of the three IDOH determinants: Relationships, Health Systems, and Cultural Continuity. Themes and corresponding subthemes are presented in Table 2. Since the intention of this cross-Native nation analysis was to identify experiences endured by many Native people in urban and rural locales in Arizona, not the experience of one specific nation, themes and subthemes were incorporated in this analysis only when the concepts or feelings were shared by TKHPs from all nations. Participant quotes (verbatim) from each of the participating nations are provided to illustrate each theme and subtheme.

### 3.1. THEME ONE: Disruption to Relationships

One of the significant threats to harmony was the disruption to relationships caused by the pandemic and tribal government orders to isolate and limit in-person social interactions. Two main subthemes were identified for disruption to relationships: (1) Fear of Social Engagement and (2) Loss of Connectivity.

#### 3.1.1. Subtheme 1.1: Fear of Social Engagement

For many of the TKHPs, the state of the pandemic brought uncertainty and fear, ultimately contributing to social disruption. With limited knowledge and understanding of the virus, and the rising rates of infection, the idea of social engagement was met with distress and anxiety. As healers, TKHPs from the three different Native nations struggled with concerns of their own, including their families’ risk and vulnerability to infection. While TKHPs wanted to engage with people, they were hesitant due to unfamiliarity with the dangers of the virus.

The following statements were made by a TKHP in each of the 3 Native nations.


*We have gone through a period where we are even afraid to shake hands as though people are evil. We are afraid of one another. It should not be like that. We were told to always be kind to one another and follow the teachings of [kinship] and love one another. But we appear to have deviated from those teachings, and we don’t know how to correct that.*


*There were some objections to some of what we went through…because they felt like it wasn’t safe yet. And then there’s other people’s [questions] like, ‘why are we so afraid of this? This is our teachings. We’re going to get through this being resilient…This is what we’re supposed to be doing, and we shouldn’t be afraid of the sickness.’ Those were some of the comments I remember hearing, and then there’s some that were in between. Yes, we’re supposed to do this, but yes, it’s a dangerous sickness, so you never know what could happen. So, we still have to be cautious of how we do* [cultural practices].


*In the beginning, it was really tough for me because I have a really hard time sitting still…I really like to travel. I really like to go anywhere, be out of the house… But on the other hand, going to the grocery store gave me anxiety. I never had anxiety in my life, about anything. I’ve been able to handle everything. But just being in the stores and around people, it’s really different. It’s a big struggle for me.*


#### 3.1.2. Subtheme 1.2: Loss of Connectivity

TKHPs described the overwhelming loss of social connectivity during the pandemic, such as loss of interactions with others in the community, including family, friends, patients, and fellow TKHPs, and a disconnect from the natural environment. Many TKHPs mentioned social isolation and the alarming COVID-19 death tolls as major disruptions. Other participants spoke of how the loss of connectivity disrupted the development of social norms for younger generations. The nuances of this subtheme are evident in these quotes.


*When you socialize with your relatives you get to see and talk with them. Before the pandemic, we had… ceremonies where people were invited to come over and that is what we got used to. And we got used to going to stores to see and visit relatives, but the mandates of our tribal government to stay home, not gather has caused fear. The thought of visiting relatives is no longer available, and we miss socializing and we are not the only one probably, the whole world is like that.*



*My grandfather always told us that to never forget the corn, he said if we forget the corn, starvation will come. The rains won’t come because there’s no corn to visit. He told us that already. I felt that we were almost in that stage, where people were dying. We’re losing family and relatives one by one, in close proximity. That was so hard to take.*



*So those are kind of some of the things, especially youth, like my son. And I want him to be able to participate in these kinds of things because he’s young and he needs to be exposed to it. But because of everything that’s going on, he really hasn’t. So that’s one of the biggest negative impacts that are really young children, even teenagers and kids in elementary. They’re missing out on who they are.*


*But here in* [participating community] *we’ve had almost 40 deaths, and in a small community where we all grew up with each other and we know their grandparents and we know their children like, those affect us. We see that and we see the reality of what this pandemic is and how it can really affect your people and so it’s been a scary time.*


*I miss that, I miss being able to connect with these other women in our community and to sit and talk and show them how to pick things, what to look for, and just that connection.*


### 3.2. THEME TWO: Disruption to Health Systems

The second threat to harmony was the disruption to Indigenous and non-Indigenous health systems, evident in nations located near or more than 100 miles from trauma centers that could manage the secondary complications of COVID-19. With high rates of COVID-19 cases, hospitalizations, and deaths, TKHPs and the Native nation’s governments worked to keep people safe. Two subthemes were identified: (1) Tension and Efforts to Integrate Systems and (2) An Unknown Threat.

#### 3.2.1. Subtheme 2.1: Tension and Efforts to Integrate Systems

TKHPs recognized the absence or limited integration of Indigenous health practices in non-Indigenous health systems and recalled the need for traditional healing. The integration of the two health systems was further complicated by COVID-19 mandates, which restricted traditional methods of healing and connection. TKHPs described the difficulties they experienced and how they addressed tensions by continuing to provide their services.


*When some of the patients got sick, they were taken to hospitals and eventually down to Phoenix. Family members and relatives were not allowed to visit. But ceremonies were done on their behalf, and they got well, and some were not able to come back. Those who did not make it back were few. Most of the patients who came back had ceremonies and they were in [locations of participating communities] on ventilators. Some of those who got really sick were not able to make it back. It is better to do whatever you can when they are just getting sick.*



*That’s another disconnect, is that why aren’t we looking to our own ways to help heal our people? The health care center is a perfect example of that. A building or a place, since it’s not like a hospital, really, it’s more like a clinic. There’s already the building there, that’s something that can be converted into or include [traditional] practices, holistic approaches, [traditional] holistic methods along with other methods, like I said, the meditation.*



*Family members aren’t allowed to see loved ones when they’re in their hospital room because they’re trying to take those COVID precautions. And although we understand that it is deeply troubling, I will say for a lot of family members, especially all of them, to not be in the same room. When family members pass away, and so COVID-19 has dramatically and I will say, negatively impacted our ability to connect to one another when it comes to people passing.*


#### 3.2.2. Subtheme 2.2: An Unknown Threat

TKHPs reflected on the adjustments made due to COVID-19 to cultural teachings about adversity and protection strategies. TKHPs recognized the disruption to traditional healing practices but were firm in their belief that specific prayers and ceremonies, even if altered by the circumstances, would provide protection.


*It is dangerous to talk about the pandemic since it seems to be alive, can travel and can hear.*



*I’m looking at the changes that I have made but I will probably go back to the way it was originally performed. However, the changes made are due to what we are faced with. I was told that minor changes can be as the need arises. I was told that one should always be flexible within certain ways. The pandemic that is among us. It should be discussed as to how we should get rid of it. Prayer offerings should be made and direction or instructions should be given by medicine people as to how we should get rid of it.*


[Elders] *Tell us also that if we don’t listen, and we’re living this unbalance way of life, something will come and shake us to wake up, to make us wake up and realize that we need to get back on the right path. So, I guess we aren’t listening and following our teachings that is why this thing came upon us. Either it would come upon us in a flood, or fire, or disease and sickness, and there’s one more thing which I can’t recall what it was. And it came as disease and sickness to wake us up and this is what we were taught. We’re told and so that’s what really amazed me. But then that’s when all the emotions start coming, because all this knowledge that we had came back and I say, “amazingly, our elders were extremely knowledgeable about these teachings and were trying to teach us, but we weren’t listening and understanding them and now we are doing everything wrong.*


*I would agree that there our cultural leaders are important people because, you know, more than ever, people are trying to understand like… what if our ancestors were here? What are the protocols, you know? And so, it’s kind of just up to me, and other cultural leaders as well to kind of step forward and say, “well, this is what we believe in and it’s about being together and working towards it together.” And so, there’s this really strong aspect.*


### 3.3. THEME THREE: Disruption to Cultural Continuity

The third threat to harmony expressed by TKHPs who practiced both in their Native language or English, was disruption to cultural continuity. Two subthemes were identified: (1) Disruption to Traditional Practices and Ceremonies and (2) Disconnect with Traditional Way of Life.

#### 3.3.1. Subtheme 3.1: Disruption to Traditional Practices and Ceremonies

TKHPs described how disruption to traditional practices and ceremonies occurred with the banning of social gatherings and requirements to adhere to social distancing guidelines. The inability to engage in ceremonies was counterintuitive, given that these traditional practices were the very means for healing and restoring harmony.


*We depend on our traditional ceremonies whenever we are confronted with adversarial situations. That was the purpose of learning these ceremonies. When we encountered the pandemic, we were told not to conduct ceremonies or gather, and they closed everything. We couldn’t do anything.*



*Definitely our culture has been impacted. Definitely the ceremonies have been impacted, especially here. Because the religious leaders decided to not perform our ceremonies to keep everyone safe… it affected my grandchildren all the way up to my parents. Everybody was affected because we couldn’t hear and see the [cultural figures]. We couldn’t go and gather for [cultural dance]. We couldn’t be at the [ceremonial space]. We couldn’t go to social dances. We couldn’t be there for weddings or for baby-naming’s. It affected everything. It gave us a glimpse of what our future can be without our ceremonies. The lawlessness that came about, nobody stepping in to help the elders.*



*Losing out on the opportunity to practice some of our traditions. Yeah, that’s probably like the biggest thing. Just because we’ve had to social distance. So, I feel like even though the tribe has put restrictions on gathering and stuff like that, a lot of people within our community abided by those and didn’t have gatherings, traditional gatherings included.*


#### 3.3.2. Subtheme 3.2: Disconnect with Traditional Way of Life

TKHPs drew a parallel between the disruption in cultural continuity and a disconnect with the traditional way of life in both urban and rural areas. Some TKHPs acknowledged that sickness in their community stemmed from abandoning cultural practices and traditional ways of living. Others described how COVID-19 restrictions prevented them from engaging in traditional ways of living that might have helped prevent sickness during the pandemic.


*I called all of the relatives together and we talked about what our elders did. Things in the presence of the unknown. I said, ‘we all seemed to have forgotten our traditional medicine, our way”.*


*And that’s why I’m saying if we were taught the proper way of our culture, it wouldn’t be that way. Because all of that is taught within our culture. You look you see people sick. There’re so many sick people with diabetes, high blood pressure, all kinds of illnesses. And it’s because we’re not taking care of ourselves. The way we should, and that also is taught, should be taught because that’s a part of it. And that’s what I see, that’s why I say that we could be a lot stronger if people understood the meaning of* [our culture].


*I think the other part about it is, this being [Indigenous] and being culturally ingrained in that part. And that’s one of the things I would say about our community. And I can see how it affected our community. Our people had a longing for ceremony, whether it be social dance or whether it be for societies or… anything, coming together, women society, men society, and especially for our [cultural figure]. People had a longing for that, and I think along the way they forgot.*



*Then going to the river and being able to harvest foods. They said, ‘oh well, the river is closed but you can still have access to cultural practices.’ But that was a little skewed in a way… In order to conduct a traditional practice, you’d have to get authorization from the police department… So then we’re allowing this non-Indigenous entity within our community to basically pass judgement on the validity of those cultural practices… And then if they did authorize it, then it was a time that they deemed okay for them… Isn’t necessarily conductive to the time that you needed to for those things… That really sucked for me because I go to the river and I practice these things, I go out and check on the plants to see when they’re ready.*


### 3.4. THEME FOUR: Recognition of the Importance of Relationships

TKHPs were first responders mitigating and assessing the threat of the COVID-19 pandemic. As healers and protectors, TKHPs had a vital role in restoring balance and harmony and addressing the community’s immediate needs. Of the IDOH determinants, the recognition of the importance of relationships was an essential goal to restore, empower and promote the well-being and resilience of patients and the community. Two subthemes were identified: (1) Continued to Provide Support/Prayer to Others and (2) Gaining Strength from Relationship with Environment.

#### 3.4.1. Subtheme 4.1: Continued to Provide Support/Prayer to Others

TKHPs described their commitment to providing support and prayer through alternative means in response to social distancing protocols. TKHPs used phones and online platforms to adjust how they communicated. Their continued communication offering support and wisdom was unwavering, as TKHPs provided encouragement, prayers, guidance, and humor to promote wellness and restore balance. TKHPs held ceremonies, said prayers, and even danced on behalf of their patients “in absentia”, i.e., their patients were not present, but the completion of the practice was reported over the phone.


*I told the people I can have mixed herbs that I can share with you. I went to their homes and told them I am going to place the herbs at your doorstep. Some of the people were really hurt by the loss of relatives. I also took food out to them and talked with them following social distancing. I tried to cheer them up and when they laughed, after joking around, and I knew they were going to be okay.*



*Those that I helped, I talked to them the same way. Even from far away, they would call me, help me during this time, this is what’s going on with me, this is how I need help, and some got affected by the pandemic. They asked me to help them by calling me, here at home, and I put down cinders, and I would put down my instruments, and pray for them.*



*They just said, let’s go out there and let’s start raising money or let’s start gathering food for people that need it. We got people that were getting wood for elderly. It was just all kind of things that I thought as far as what we learned is that when things happen, people step up and contribute. And so, I hope that these kinds of programs…will continue to exist, that they don’t just go away once COVID’s done.*



*I’m dancing for them because they’re not here. And like, just like how it is in general, I’m dancing for those that have gone on and you can’t be here. And I think that’s what helped me be more resilient because I want to keep doing that, even though a platform that we’re using now for virtual teaching.*


#### 3.4.2. Subtheme 4.2: Gaining Strength from Relationship with Environment

For TKHPs, the relationship with the natural world is fundamental to restoring balance and harmony. TKHPs from each community expressed their reliance on the land and the environment and the need to respect and connect with the natural world to strengthen oneself and ensure holistic well-being.


*This is how COVID-19 seems to be, but with our prayers and songs we can become rejuvenated. At the time when it first came among us, I realized that it becomes part of the air, light, earth and our environment. It becomes part of our environment, and we breathe that in. It is just like a drop of water in a pond that changes the colors and that is the way the air is. Thus, if you address the air, sun, and the earth, you can gain strength from it, and I believe that is what you can call resilience.*



*I wanted to help focus on restocking our stores of beans and corn because of the drought before and during the pandemic. We still had drought, and our stores of corn and beans were low. I knew my parents were worried about it with us being corn clan and everyone coming to us for corn and everything. I immersed myself fully in that, in helping to restore our corn and beans. Me and my children started helping with the bigger fields and that’s what I feel helped us… To remember that we are provided for if we do the hard work… that the Earth and the rains and everything will provide for us if we don’t forget where we come from, and what we should be doing.*



*The more I learn about the culture, the more I understand that there is a lot of practices, traditions, cultural ideas and even in the language, there’s a lot of words to describe things. And how…our ancestors used to use to think, because when we look at the culture where we often find that there’s this this idea of working together. This desert is a harsh environment. It’s a beautiful environment where there’s so much life, there’s so much diversity out here and there’s so much life in this desert…but that’s to say it can be harsh…and any native to Arizona will know that the summers can get brutal. And when you think about just how brutal it can be out here, working or being by yourself is going to be detrimental because you can’t really survive out here by yourself. You need help. People always need help and so when we look at our whole town and when we start looking at the art, the ancestral records, archaeological records, even the language, we do see there are those who have worked together since time immemorial, pretty much forever since the beginning of time. And we need to work together to survive. And so, there’s a very strong and inherent connectivity that connects us all.*


### 3.5. THEME FIVE: Recognition of Indigenous Health Systems

TKHPs recognized that healing occurred in traditional and non-traditional health settings. However, in many cases, community members sought out TKHPs for health and healing during the pandemic, recognizing the critical need for Indigenous health systems and traditional medicine to maintain harmony and balance. Three subthemes were identified: (1) Responsibility to Patients, (2) Self-care, and (3) Recognizing the Strength of Indigenous Healing Practices.

#### 3.5.1. Subtheme 5.1: Responsibility to Patients

In reflecting on harmony and balance, TKHPs highlighted their responsibility to connect with and support patients, even if the connections were happening virtually. Several TKHPs acknowledged overcoming their challenges but maintained resiliency and focused on meeting patients’ needs.


*As I walked down the hall of the hospital, I saw many of my relatives needing help. People were coming around wanting to know if I was helping the people. I finally said I can help. It is up to us. Many people whom we have depended on us, left us. Many have been affected at this time. In the past this was not the case, but now we have to do whatever to help ourselves.*



*People were calling saying “I need prayers” that is what happened. So, so this happened to them. We were busy and then having to talk to them, to keep the faith. Even down to telling them, don’t just sit around, and don’t just watch TV in your house, to stay physical and so we had surprisingly a large number of people wanting prayers. In a way we didn’t have time to fully immerse in being frightened.*



*And so it was difficult, it was tough but at the same time, I think being [traditional] you tend to lean on prayer and trying to have positive thoughts and saying okay we’re still taking care of the people, we’re still thinking of the people and so when things did ease up a little bit, there have been times where I’ve been able to gather with the [ceremonial groups] again and it was good to see.*



*I would just say, the last thing I would just add is that being [identified the specific Native nation] gives me a different perspective on this whole sickness in general. It gives me a different perspective of how I would view it in terms of just a regular person in terms of like, different from the United States, different from the city, different from other and even another Native American or Indigenous tribe. [identified the specific Native nation] has given me a very strong impression of a very positive one where I feel connected to my people, even though we’re meeting virtually.*


#### 3.5.2. Subtheme 5.2: Self-Care

TKHPs described how maintaining Indigenous health practices by enacting self-care initiated a path of recovery and resilience. Practicing self-care is a personal responsibility that TKHPs honor to create harmony and balance within oneself to empower and care for others.


*That is why you get up early and run in the morning. Make yourself experience it and work. And also, you’re thinking you think good things and then make sure you plan good. This way some sometime in the future when you experience this [adversity] you will be strong and resilient toward it.*



*The other thing is just going out and praying in the morning and praying for everybody’s well-being and not to be affected by COVID to protect everyone from it. And we are praying for them to be here for a long time. And that was mainly my prayers, to not let this sickness get to them and to live a long, happy, healthy, strong life and humbly reach old age. In the morning to go pray outside. And so that was those were the things that kept me going. I mean, that’s what I did to help me. A lot of prayer. And even in the house I talk to, there’s grandpa’s pictures, there’s grandma’s picture in there. And I know that in the spirit world. So, if I talk to them to help all of us and to watch over and protect our children and grandchildren, not let this sickness get them so they will live a long life. Even to them I prayed to their pictures.*



*I think many of our people have carried on their beliefs pretty much how I indicated earlier, practicing at an isolated level to carry out those prayer and practices and physical activities to maintain our way of life. I continue to do that, but I don’t really know if the other spiritual leaders have done the same thing but if we are taught that we, no matter what to keep doing this, even though we see ourselves not with a group.*



*I know to pray, I prayed. Kind of forced myself to sit at home, forced myself to be home, not be out and just… Even like that, taking my previous answers and applying it to this one that like, well, there’s a reason why, even for me, like as much as I have a hard time sitting still, maybe I do need to just take this time and just be home and be reverent.*


#### 3.5.3. Subtheme 5.3: Recognizing the Strength of Indigenous Healing Practices

TKHPs practicing in urban and rural, and large and small nations, reflected on their strong sense of identity as healers and wisdom keepers, which is a testament to their ability to draw strength from culture and tradition. TKHPs recognized that their ability to remember and maintain cultural teaching and practice, reinforced their spiritual strength. Furthermore, they noted that the pandemic influenced revitalization and recognition of the power of Indigenous health practices.


*They said these were the things our elders did at one time when an illness came among us. I said why are we just talking about it. Why can’t we do something similar? We did this by putting herbs into the fire and smoked our livestock, corrals, [houses], vehicles, and ourselves. We gave some of the herbs and told people to smudge all things at their homes. We told them to boil some of these herbs and drink them. We were told that this is what our people did because of the belief that the sharpness from some of these herbs were used against the different illness that are out there. I did ask some of the medicine men what type of herbs should not be burnt and they told me the only one… because this is a sacred plant.*


*There were several medicines that was brought back into use, such like sage. The horse were made* [to be eaten as] *meat again. These were our vaccine long ago.*

*The spiritual prayers are strong enough to overcome that pandemic. So sometimes I think we take challenges too, also some criticism, some misunderstanding within their own families when we did things that may not be as closure with the policies that are being enforced by the tribe or by health institutions to carry out some of our beliefs, but to maintain the* [traditional] *way of life, sometimes we have to take it in our own spiritual strength to do it. And expecting that this would help us to overcome this pandemic and sometimes we are misunderstood thinking that maybe we are being selfish instead of trying to really make the people understand that we’re doing it because this is the power of our way of life to deal with problems like this, but it was a big challenge in that way.*


*We have traditional beliefs, we have our faith, I guess we call it. However, it is that you see it, that’s really helped. I think that grounded me, and one of the observations and just thinking on my own.*


### 3.6. THEME SIX: Restoring Cultural Continuity

Despite encountering hardships, TKHPs described their efforts to sustain cultural practices and their methods of adaptation during the pandemic. Three subthemes emerged: (1) Continued Use of Ceremony and Medicines, (2) Return to Teachings, and (3) Use of Technology.

#### 3.6.1. Subtheme 6.1: Continued Use of Ceremony and Medicines

To ensure cultural continuity in the face of COVID-19, TKHPs described how they and their community members continued conducting ceremonies and using traditional medicine. Individuals offered prayers at sacred spaces in their community and gathered traditional medicines. Others described engaging in their own versions of ceremony and medicine within their immediate family unit.

*We depended on the offerings of the sacred mineral stones. We went to* [sacred mountains] *…and offered protection prayers. There were several of us doing that and many medicine men also did that elsewhere. We also gathered medicinal herbs and used that with our families and relatives. This is what we did throughout the pandemic. We used ceremony and herbs throughout the pandemic.*

*I will continue to practice what I was taught to keep on and sustain the* [Indigenous] *way of life traditionally, spiritually or physically. I was able to continue those things at an isolated level, like at home rather than at gathering. And to keep that going strong, even though we were restricted from group gatherings. And I think it brought about a big challenge to keep it going and to make sure that our* [Indigenous] *way of life continues despites the pandemic that we’re facing.*

*Our ceremonies are our way of healing ourselves. This is what I learned. Because we couldn’t do those things during* [ceremony]*, when we didn’t get our feathers, it was so sad. My father told us that we always acknowledged that day anyway, and we did. We gathered as a family and we didn’t get our feathers, but we went as if we got our feathers prayed in the same way we would pray during* [ceremonial] *time. We all still gathered during those significant times. We had traditional meals, because that’s what my dad wanted and so we all prepared it. We got ready just like it was going to be what we do. So, we still gather in that way. I felt that kept us together, stronger as a family. We continued to pray in those kinds of ways.*

*My Indigeneity, how me being* [Indigenous] *really helped me, it was because, like I fell back on a lot of those underlying principles and ways of being. Even at the time that it was scary to do, because it was scary, people needed help and I don’t want to get sick, and I don’t want my kids to get sick, but we can do it in a good way. We figured out ways to make it work.*

#### 3.6.2. Subtheme 6.2: Return to Traditional Indigenous Teachings

Participants also described the importance of returning to teachings to ensure cultural continuity. They described the need for the community as a whole to return to teachings to bolster resilience. Individuals also noted the importance of returning to teachings for mental well-being. Additionally, TKHPs noticed that younger generations were seeking guidance on cultural teachings and practice, which speaks to the intergenerational nature of cultural continuity. There was a renewed interest by the youth in the cultural teachings and practice


*We notice that a lot of the young people came to rekindle an interest in traditional ceremonies and this medicine. ‘How do you use the herbs?’ they said. ‘How do you use this sage? Is there a particular way of going to gather them and is there a prayer you say? What kind of prayer do you do?’ It’s these things, so if there is anything there was a renewed interest by young people.*


*I think because of our teachings. We are resilient. I mean, we’re taught in our culture how to survive. So, if people really paid attention to that, our* [Indigenous] *people. You know, we wouldn’t really worry so much about survival if we practiced it more… I believe [our Indigenous Community] has a really strong culture that is geared towards resilience. I believe we are resilient and if we continue practicing our culture, we’d be even stronger… A lot of the teachings aren’t being taught the way they should be and that’s why you see a lot of sickness going around. And that’s where we need to focus on, teaching what we’re doing and why we’re doing it. And we’d be so much stronger.*


*During this last year, when I can help people when we were doing things like that really helped me, grounded me and made me really feel like part of this community and that I was doing the things that I was taught to do, that my parents taught me to do, that their parents taught them to do. Connected me all the way through the service to our ancestors.*



*With that kind of thinking, I started to apply that to today with what we’re going through with this whole COVID-19 sickness. Thinking back, a long time ago when we had these stories or even like the floods that happened, they say something took place first. The people were doing something that they weren’t supposed to be doing, or they were either weren’t doing something, or they were doing too much. But whatever it was, it caused an imbalance. Then from that imbalance, there was a reaction, whether it be a flood or sickness, something like that. So, then I started thinking about that, how that even could apply to this and with a lot of our traditional ways.*


#### 3.6.3. Subtheme 6.3: Use of Technology

COVID-19 forced many changes across the globe. This adaptation was also true for TKHPs in our study. Participants described the use of technology as a facilitator of cultural continuity in Indigenous communities. TKHPs and their communities used technology to teach language, history, and cultural practices. Technology was also used to facilitate prayer, some ceremonies, and virtual social gatherings.

*One of the changes is doing prayers by tele-conference because of the situation where the person may be at another location. By doing it through tele-conference, the patient and their relatives can hear the prayers. This makes them strong and is a change. In our cultural ways, you have to be physically present for prayers and ceremonies. However, you can’t do that for* [Ceremony] *only protection prayers. The way it has been done in the past is if a service man is in Germany and is suffering from some illness a relative can sit in for that person. The prayers are effective, and the person gets well and that is the way it has always been done. The air covers the whole world and even if the person is in Iraq or Russia, it will still get to him. So, prayers can be said for that person as if he/she were sitting here.*


*It has changed us because I’ve been able to keep in contact through our technology, mainly through the phone, with our spiritual leaders. Keeping in contact and communicating the importance of our activities and also keeping informed about how they’re carrying on their practice. So, we’re all working together even though we’re away from one another. We’re still carrying on the prayers that we normally do for our people, for us and to be able to kind of use technology to carry on the ceremonies and teachings.*


*Now we have this virtual and social media to kind of push the culture…When before* [COVID-19] *again, we didn’t have it [technology] fully, but it was there. But now that we kind of needed it, we had to put in place that infrastructure so that we can do that. It has been successful because our multimedia team monitors our social media posts, and they monitor activity. And you know, we have a lot of positive feedback from our posts. It kind of goes to show that people were waiting for stuff like this to come out for a while. And even though COVID-19 was, you know, we rather have not had happened, this was probably the biggest positive.*

## 4. Discussion

In the US, some health care facilities serving predominantly Native American patients have integrated TKHPs into clinical practice for individual patients who request the service [26,27]. Western medicine has not consulted with Indigenous TKHPs in times of community-wide crises, internationally and in the US [28]. Misinformation about Indigenous Peoples’ health risks and resilience in the face of adversity can be perpetuated by contributing factors such as the absence of TKHP voices, non-Indigenous scholarship, and colonial bureaucracies [29,30,31]. This study presents TKHP voices from three different Arizona Native Nations, to represent the role of Indigenous knowledge in maintaining community health through social and cultural connections. As such, these narratives reinforce previous work that unveils the significance of including Indigenous knowledge, wisdom, and skills from TKHPs to address public health emergencies in Native American communities [32,33,34].

For TKHPs, the context of the COVID-19 pandemic was not only a physical illness, but also, a social and cultural disruption. The pandemic-related fear and vulnerability were consistent among Indigenous populations, with rising concerns about the loss of Elders and traditional knowledge and languages [35]. The COVID-19 virus was even portrayed as a “merciless monster” that threatened the health and livelihood of tribal nations [36]. The pandemic created disharmony by disrupting Indigenous ways of living. Studies indicate that the interruption to culture contributes to poor psychological resilience, resulting in an increased risk for physical, mental, psychological, spiritual, and environmental health challenges [7]. While the TKHPs interviewed from the three Arizona Native nations shared varied perspectives about their role throughout the pandemic, many articulated this process of disharmony caused by the global pandemic and the need to restore and regain harmony through the use of traditional knowledge and practices. Recognizing the crisis created by the pandemic and the urgency to sustain traditional knowledge, Indigenous communities throughout the US made efforts to revitalize cultural practices [35,37]. The connection to traditional practices, knowledge, and languages is imperative to promoting psychological resilience and is recognized as protective health protective [7].

The themes of disruption to harmony and restoration of harmony that emerged from the interviews with the TKHPs reflect the persistence and resilience of TKHPs in upholding their traditional knowledge and practices, affirming the passing of knowledge to the next generation and ushering in peace and healing after unprecedented loss and hardship. TKHPs provided a sense of hope that the pandemic would not be long-lasting and that traditional medicine and practices could help alleviate the pain of the pandemic.

This study revealed the value of TKHPs in communities during crises, as exemplified by the COVID-19 pandemic. Indigenous worldviews and systems of relationality are deeply rooted in the interconnectedness of the material and spiritual worlds, where TKHPs are the collective healers, protectors, and safeguards of Indigenous wisdom and ways of life [2,5]. This study reflects holistic recognition of well-being that is also echoed in international Indigenous communities [38]. Findings from this study support the need to amplify TKHPs teachings and knowledge in times of community-wide destabilization.

### Limitations and Strengths

A limitation of this study is the small number of interviewees, with 22 TKHPs representing three Native nations. Additionally, this study partnered with only three Native nations in the US Southwest; hence, the results may not be generalizable to the entirety of Indigenous TKHPs. Another potential limitation is that some of the interviews were conducted in Native languages, and the translation of words into English may have been altered, thus, not conveying a representation of their true meaning.

While these limitations should be considered, the research team was committed to ensuring that every phase of this project recognized and involved members from each community. Our project’s large, primarily Indigenous, research team made concerted efforts to build and support trust with leaders and citizens within each nation. Furthermore, the research team ensured that the participants’ stories and lived experiences were analyzed, interrupted, and reviewed by Indigenous team members, including those from the participating Native nations.

## 5. Conclusions

TKHPs’ role was influential in guiding and supporting citizens of Native nations in Arizona to heal and maintain their mental and spiritual health throughout the COVID-19 pandemic. Their contributions to their communities’ physical, social, cultural, and spiritual health during the COVID-19 crisis elucidates the need to ensure that they are consulted and included in emergency response teams during future public health emergencies and the development of safety protocols. Their traditional knowledge, teachings, songs, and prayers are foundational assets in Native American communities, offering invaluable lessons to youth and promoting mental wellness and resilience among the members of their communities. TKHPs’ holistic approach to addressing COVID-19 and pandemic-related challenges extended beyond the typical Western approaches to medicine, making them vital stakeholders and providers for current and future efforts in improving the health status, well-being and resilience of Native American communities.

## Figures and Tables

**Figure 1 ijerph-22-01432-f001:**
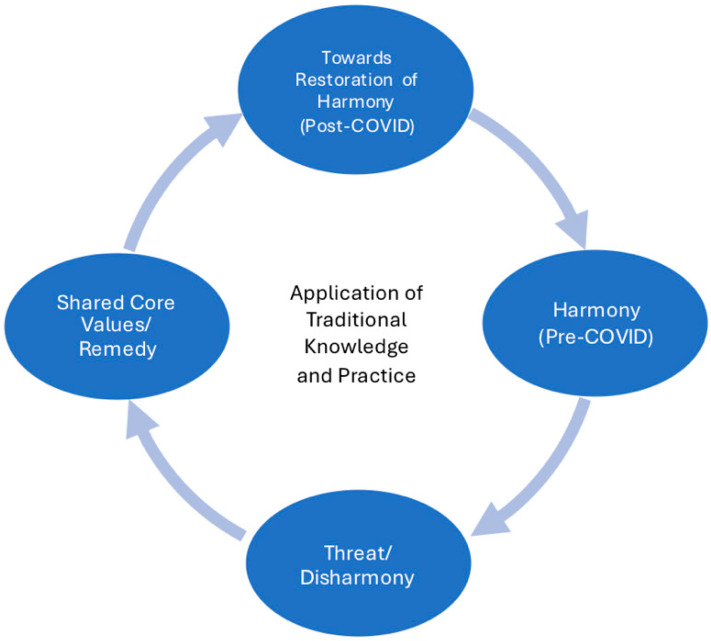
Application of Traditional Knowledge and Practice from Harmony (Pre-COVID) to Restoration of Harmony (Post-COVID).

**Table 1 ijerph-22-01432-t001:** Demographic Characteristics of Traditional Knowledge Holders and Practitioners interviewed (*N* = 22).

**Gender**	Female	6 (27%)
	Male	16 (73%)
**Age Range (yrs.)**	25–34	2 (9%)
	35–44	5 (23%)
	45–54	1 (4%)
	55–64	3 (14%)
	65+	11 (50%)

**Table 2 ijerph-22-01432-t002:** Themes of Disruption and Restoration.

**THEME ONE—DISRUPTION TO RELATIONSHIPS**
Subtheme 1.1: Fear of Social Engagement Subtheme 1.2: Loss of Connectivity
**THEME TWO—DISRUPTION TO HEALTH SYSTEMS**
Subtheme 2.1: Tension and Efforts to Integrate Systems Subtheme 2.2: An Unknown Threat
**THEME THREE—DISRUPTION TO CULTURAL CONTINUITY**
Subtheme 3.1: Disruption to Traditional Practices and Ceremonies Subtheme 3.2: Disconnect with Traditional Way of Life
**THEME FOUR—RESTORATION OF THE IMPORTANCE OF RELATIONSHIPS**
Subtheme 4.1: Continued to Provide Support and Prayer to Others Subtheme 4.2: Gaining Strength from Relationship with Environment
**THEME FIVE—RESTORATION OF INDIGENOUS HEALTH SYSTEMS**
Subtheme 5.1: Responsibility to Patients Subtheme 5.2: Self-care Subtheme 5.3: Recognizing the Strength of Indigenous Healing Practices
**THEME SIX—RESTORATION OF CULTURAL CONTINUITY**
Subtheme 6.1: Continued Use of Ceremony and Medicines Subtheme 6.2: Return to Teachings Subtheme 6.3: Use of Technology

## Data Availability

The data sets generated and analyzed during this study are not publicly available because they include data from participants of sovereign Native nations, and these data cannot be released without explicit permission from each nation. Please feel free to contact the principal investigator, Julie Baldwin, if you have questions.

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
