# Peer review of "Traditional Knowledge Holders and Practitioners: First Responders in Native Nations During the COVID-19 Pandemic"

_ijerph, 2025, doi:10.3390/ijerph22091432_

Round 1
Reviewer 1 Report
Comments and Suggestions for Authors
This manuscript does an excellent job in clearly demonstrating how traditional practices and spirituality played an important role in the health and wellbeing of 3 Native communities in present day Arizona during the COVID-19 pandemic. The research provides vibrant results that may advance the social determinants of health framework to include the three proposed Indigenous factors: (1) relationships with all living beings and the natural environment, (2) the intersection of non-Indigenous and Indigenous health care systems, and (3) cultural continuity. In my opinion, it would be even stronger by restructuring the results section and amending a few small pieces, as described below.
General comments:
In the Results section text and quotes there is so much overlap in themes and content that it makes reading the results and seeing clear delineations between the themes difficult. The 3 overarching themes are strong, why not explore the 3 themes (which include both disruption and restoration) instead of breaking them apart into disruption and restoration/recognition? For example, the quote on lines 507-511 has both disruption and restoration, as do so many other quotes. By combining the 6 themes into 3, you could illustrate the continuum of harmony-discord/ disruption-restoration, and the integral roles TKHPs play in moving toward restoration (harmony) when experiencing disruption (discord). Combining would allow the quotes to flow better –tell a story—rather than feel like disassociated snippets.
Specific comments:
- It would be useful to add a sentence about how the community partners determined who was a TKHP because the explanation will add additional validity to the sampling design for people not familiar with Indigenous communities.
- In Figure 1, I wonder about “restoration” of harmony. I realize that simplification of terms is necessary for a figure; I am thinking about how in some teachings, harmony-discord is not an either/or, it is a continuum in which it there is never entirely one or the other. Perhaps it is different for the communities in which you did this work. I do feel like “toward” harmony (restoration) is more fitting.
- Put a space between each quote. It is difficult to discern if each new paragraph is a new quote or the continuation of a quote from the same person
- The themes in Table 2 are called disruption and restoration but themes 4-6 are called recognition and not restoration. It seems like restoration is a more fitting term—why choose recognition? Suggest changing it back.
Reviewer 2 Report
Comments and Suggestions for Authors
Thank you for the opportunity to review your article. The manuscript does well to center the voices and experiences of traditional knowledge holders during the COVID-19 pandemic.

Round 2
Reviewer 2 Report
Comments and Suggestions for Authors
Thank you for your clarifications and edits. I am excited to see this important work being shared widely.